# BM-CL: Bias Mitigation through the lens of Continual Learning

## Abstract

Biases in machine learning pose significant challenges, particularly when models amplify disparities that affect disadvantaged groups. Traditional bias mitigation techniques often lead to a *leveling-down effect*, whereby improving outcomes of disadvantaged groups comes at the expense of reduced performance for advantaged groups. This study introduces Bias Mitigation through Continual Learning (BM-CL), a novel framework that leverages the principles of continual learning to address this trade-off. We postulate that mitigating bias is conceptually similar to domain-incremental continual learning, where the model must adjust to changing fairness conditions, improving outcomes for disadvantaged groups without forgetting the knowledge that benefits advantaged groups. Drawing inspiration from techniques such as Learning without Forgetting and Elastic Weight Consolidation, we reinterpret bias mitigation as a continual learning problem. This perspective allows models to incrementally balance fairness objectives, enhancing outcomes for disadvantaged groups while preserving performance for advantaged groups. Experiments on synthetic and real-world image and tabular datasets, characterized by diverse sources of bias, demonstrate that the proposed framework mitigates biases while minimizing the loss of original knowledge. Our approach bridges the fields of fairness and continual learning, offering a promising pathway for developing machine learning systems that are both equitable and effective.

## 1 Introduction

Machine learning systems have achieved remarkable success in a variety of tasks, ranging from automatic translation to facial recognition. However, as these technologies are increasingly deployed in society, concerns about bias and discrimination have emerged (Buolamwini & Gebru, 2018; Mehrabi et al., 2021). Biases often manifest as performance disparities between demographic groups, undermining the reliability and fairness of these systems, especially in sensitive domains such as healthcare, finance, and public policy.

Such disparities can arise from various sources, including data imbalance, label noise, spurious correlations, or intrinsic characteristics associated with demographic groups (Zong et al., 2022). Among these factors, data imbalance is one of the most common (Larrazabal et al., 2020), as many datasets often lack demographic diversity. Spurious correlations, on the other hand, arise when models exploit irrelevant features as predictive signals (Izmailov et al., 2022). Existing bias mitigation techniques, while effective in tackling group disparities, tend to suffer from the *leveling-down effect*, whereby performance improvements for disadvantaged groups negatively impact the performance of advantaged groups (Zietlow et al., 2022; Mittelstadt et al., 2023). This trade-off highlights the need for innovative approaches that promote fairness without compromising overall performance.

Continual learning, a paradigm enabling sequential learning without forgetting prior knowledge (Chen & Liu, 2018), offers a promising direction to address these challenges. We hypothesize that the leveling-down effect can be interpreted as a form of *catastrophic forgetting* (French, 1999), where optimizing for disadvantaged groups leads to a loss of knowledge about advantaged groups. To tackle this issue, we introduce *Bias Mitigation through Continual Learning* (BM-CL), a bias mitigation strategy inspired by continual learning techniques such as Elastic Weight Consolidation (EWC) (Kirkpatrick et al., 2017) and Learning without Forgetting (LwF) (Li & Hoiem, 2017). BM-CL follows a two-step training process specifically designed to correct a biased model against

disadvantaged groups without compromising outcomes for advantaged groups or affecting overall performance.

We validate our approach on both synthetic and real-world datasets widely used in bias mitigation research, demonstrating its effectiveness across diverse scenarios. Compared to baseline bias mitigation techniques, our method consistently enhances performance in disadvantaged groups with minimal leveling-down. By framing bias mitigation as a continual learning problem, our work paves the way for leveraging the extensive toolkit of existing continual learning methods to address fairness concerns.

## 1.1 RELATED WORK

In the last decade, a plethora of studies have demonstrated that machine learning systems can exhibit biases against specific demographic groups, often defined by protected attributes such as gender, age, or race (Angwin et al., 2022; Buolamwini & Gebru, 2018; Larrazabal et al., 2020; Seyyed-Kalantari et al., 2021). These findings have prompted growing awareness in the research community about the need to not only enhance accuracy but also improve fairness in decision-making outcomes.

Group fairness is among the most widely used definitions of algorithmic fairness in the literature (Dwork et al., 2012), which aims to reduce inequity in decisions across groups defined by protected attributes. In the context of binary classification tasks, fairness techniques often strive for group parity using specific metrics. Examples include demographic parity (Wachter et al., 2021) or equal opportunity (Hardt et al., 2016). Alternatively, minimax group fairness (Diana et al., 2021) focuses on reducing the worst-case outcomes, ensuring that the group facing the greatest disparity is treated as equitably as possible. Recent studies (Zietlow et al., 2022; Mittelstadt et al., 2023; Ferrante & Echeveste, 2025) highlight a trade-off in these approaches: many current techniques for enhancing group fairness often do so at the cost of reduced performance in advantaged groups, i.e. those whose initial outcomes already exceed the average. This phenomenon, known as *leveling down*, presents significant risks for machine learning technologies, particularly in critical scenarios like healthcare (Ricci Lara et al., 2022), where it is ethically imperative to ensure that fairness interventions do not compromise the quality of care for any group. Forcing fairness through leveling down may result in models that, while appearing fair by reducing group differences to nearly zero, are equally harmful to all groups (Mittelstadt et al., 2023; Sabuncu et al., 2025). In contrast, our work aligns with the principle of *positive-sum fairness*, recently proposed by Belhadj et al. (2024), which seeks to improve outcomes for disadvantaged groups without sacrificing performance for advantaged groups. Here, we demonstrate that this goal can be achieved by reformulating the performance decrease in advantaged groups as a forgetting problem within the context of continual learning.

Continual learning (CL), also known as *incremental* or *lifelong* learning (Chen & Liu, 2018) is a paradigm that aims to mimic the human ability to learn continuously and adapt to new situations. A major challenge in this field is mitigating *catastrophic forgetting* (French, 1999; Li & Hoiem, 2017), where performance on prior tasks deteriorates when learning new ones. Our work investigates the leveling-down phenomenon in bias mitigation strategies as a form of catastrophic forgetting. To address forgetting in classification tasks, two primary approaches exist in CL literature: data-based techniques and prior-based techniques (De Lange et al., 2021). Data-based methods extract and transfer knowledge from a prior model to a new model trained on new data. One example is the LwF method (Li & Hoiem, 2017), which uses the predictions of the previous model as pseudo-labels for future tasks, avoiding the need to access previous task data when incorporating new ones. Prior-based approaches, on the other hand, estimate a distribution over model weights, which serves as a prior when learning with new data. Among these, EWC (Kirkpatrick et al., 2017) uses the Fisher information to identify the model parameters critical for solving previous tasks.

While CL methods have demonstrated success in adapting to new tasks over time, their application to bias mitigation remains an underexplored area. The few studies in this intersection include Churamani et al. (2022), which proposes a domain-incremental continual learning approach to reduce bias in facial expression and action unit recognition. This method allows models to adapt to new domains while maintaining fairness across demographic groups, but limits its exploration to regularization-based CL approaches as well as naive rehearsal. More importantly, it does not augment these methods with strategies to improve fairness. More recently, Bayasi et al. (2024) in-

Figure 1: Overview of the proposed BM-CL framework, composed of two-stages. First, we train a model in the standard way (by ERM) and identify best and worst groups ($\mathcal{G}_{best}$ and $\mathcal{G}_{worst}$). Then, we fine-tune the model using bias mitigation to improve performance on $\mathcal{G}_{worst}$ and continual learning to preserve performance on $\mathcal{G}_{best}$.

troduced *BiasPruner*, which combines continual learning with bias mitigation by pruning neurons that contribute to learning spurious correlations, thus enhancing fairness in neural networks.

Building on these insights, we propose BM-CL to address the challenge of creating fair models by integrating task-incremental continual learning principles with bias mitigation strategies. Specifically, we show that by combining bias mitigation with LwF, we can reuse model predictions from advantaged groups in previous training stages as pseudo-labels in subsequent training to preserve knowledge for these groups, while optimizing for disadvantaged ones. Furthermore, EWC allows constraining weights that are key to performance in advantaged groups, allowing less significant weights to adjust and improve outcomes for disadvantaged groups. Our experiments show that BM-CL improves worst-group performance while consistently preventing performance degradation in advantaged groups, aligning with the principle of positive-sum fairness.

## 2 INTEGRATING BIAS MITIGATION AND CONTINUAL LEARNING

### 2.1 PRELIMINARIES

We consider a supervised classification problem where the goal is to predict a label $y \in \mathcal{Y}$ for a given input $\mathbf{x} \in \mathcal{X}$. To achieve this, we train a model $f(\mathbf{x}; \theta) : \mathcal{X} \to \mathcal{Y}$, which is parameterized by $\theta \in \Theta$, using a dataset of $n$ samples, denoted as $\{(\mathbf{x}_i, y_i)\}_{i=1}^{n}$. Here, $\mathbf{x}_i \in \mathcal{X}$ represents the input features and $y_i \in \mathcal{Y}$ is the corresponding target label.

The standard framework for training supervised learning models is empirical risk minimization (ERM) (Vapnik, 1999), which optimizes $\theta$ by minimizing the average loss over the training set. Although ERM is effective in optimizing overall accuracy, it does not ensure fair performance across subgroups, particularly in datasets with imbalances or spurious correlations (Sagawa et al., 2019), where majority groups dominate optimization and minority groups underperform.

To address this, samples are typically associated with a group $g_i \in \mathcal{G}$, where $\mathcal{G} = \mathcal{A} \times \mathcal{Y}$, with $a \in \mathcal{A}$ representing an attribute of interest. Bias mitigation methods then focus on reducing group disparities by improving worst-group accuracy. Methods like *Group Distributionally Robust Optimization* (GroupDRO) (Sagawa et al., 2019) explicitly minimize the worst-group error, ensuring that underrepresented groups are not overlooked during training. Group rebalancing methods provide another approach to bias mitigation. For example, resampling methods (Idrissi et al., 2022), referred to as *ReSample* in this work, aim to adjust group contributions by upsampling minority groups. Another resampling method is *Just Train Twice* (JTT) (Liu et al., 2021), a two-stage approach that first trains an ERM model to identify samples with high error (which are likely to belong to worst-performing groups), and then upweights them in a second training stage. In addition to these methods, *Invariant Risk Minimization* (IRM) (Arjovsky et al., 2019) has been proposed to improve robustness by enforcing invariant predictors across environments. IRM has become a widely used baseline in settings involving spurious correlations and group-based disparities.

In this work, we adopt these methods as baselines to highlight the leveling-down effect and motivate our integration of continual learning strategies into bias mitigation.

## 2.2 TWO-STAGE FRAMEWORK

We interpret the leveling-down effect as a form of forgetting, where fairness optimization degrades advantaged groups. To address this, we define BM-CL as a framework which combines bias mitigation with continual learning in two training stages (Fig. 1).

### 2.2.1 BASELINE TRAINING WITH ERM

In the first stage, we train the model to achieve high overall accuracy using the traditional ERM loss, defined as

$$\mathcal{L}_{\text{ERM}}(\theta) = \frac{1}{n} \sum_{i=1}^{n} \ell(f(\mathbf{x}_i; \theta), y_i), \tag{1}$$

where $\ell : \mathcal{Y} \times \mathcal{Y} \rightarrow \mathbb{R}^+$ is the loss function (e.g. cross-entropy loss) that quantifies the discrepancy between the predicted output $f(\mathbf{x}_i; \theta)$ and the true label $y_i$.

We empirically observed that ERM fits advantaged groups faster than disadvantaged groups, which aligns with similar observations made in previous studies (Nam et al., 2020; Liu et al., 2021). Thus, we limit the ERM training to a fraction of the total number of epochs. This fraction is controlled by a hyperparameter $\rho$, referred to as the pretraining ratio.

After training, the model is evaluated on the validation set to compute the accuracy $\alpha_g$ for each group $g$. Based on this evaluation, groups are then partitioned into two disjoint subsets: the best-performing groups ($\mathcal{G}_{\text{best}}$) and the worst-performing groups ($\mathcal{G}_{\text{worst}}$). The partition is determined by a threshold $\tau$ representing the balanced accuracy, i.e. the mean accuracy across all groups, $\tau = \frac{1}{|\mathcal{G}|} \sum_{g \in \mathcal{G}} \alpha_g$, with $|\mathcal{G}|$ denoting the total number of groups. Those with $\alpha_g > \tau$ form $\mathcal{G}_{best}$, the rest $\mathcal{G}_{worst}$. This procedure classifies the groups into those that perform above and below the average accuracy threshold, and provides the basis for the second stage.

### 2.2.2 FINE-TUNING FOR BIAS MITIGATION WITH CONTINUAL LEARNING

In the second stage, we propose to fine-tune the model using a traditional bias mitigation method (e.g., GroupDRO or ReSample), but regularizing them to avoid forgetting via CL. The goal is to improve performance for disadvantaged groups ($g \in \mathcal{G}_{\text{worst}}$) while preserving knowledge of the best-performing groups ($g \in \mathcal{G}_{\text{best}}$). We do so by optimizing the objective

$$\mathcal{L}_{\text{BM-CL}}(\theta) = \mathcal{L}_{\text{BM}}(\theta) + \lambda \mathcal{L}_{\text{CL}}(\theta), \tag{2}$$

where $\mathcal{L}_{\text{BM}}$ is the loss associated with the bias mitigation strategy, $\mathcal{L}_{\text{CL}}$ is the continual learning regularizer, and $\lambda$ determines the relative importance of the continual learning regularization compared to the bias mitigation loss. For $\mathcal{L}_{\text{BM}}$, we considered GroupDRO and ReSample as bias mitigation strategies; however, the framework is modular, so other methods could be considered. For $\mathcal{L}_{\text{CL}}$ we propose Learning without Forgetting (LwF) and Elastic Weight Consolidation (EWC) as the CL methods:

**Learning without Forgetting**: LwF employs knowledge distillation (Hinton, 2015) to retain the previous predictions of selected groups. Here, it is implemented using the Kullback-Leibler (KL) divergence (MacKay, 2003) between the current and previous predictions for the samples of the best-performing groups,

$$\mathcal{L}_{\text{CL}}(\theta) = \frac{1}{|\mathcal{I}_{\text{best}}|} \sum_{i \in \mathcal{I}_{\text{best}}} \text{KL}(q_i^*, q_i), \tag{3}$$

where $\mathcal{I}_{\text{best}} = \{i \mid g_i \in \mathcal{G}_{\text{best}}\}$ denotes the set of indices of the best-performing samples. The predicted probabilities from the previous model ($\theta^*$) and the current model ($\theta$) are computed by applying the softmax function $\sigma$ to model outputs,

$$q_i^* = \sigma\left(\frac{f(\mathbf{x}_i; \theta^*)}{T}\right), \quad q_i = \sigma\left(\frac{f(\mathbf{x}_i; \theta)}{T}\right), \tag{4}$$

where $f(\mathbf{x}_i; \theta^*)$ is the output of the previous model and $f(\mathbf{x}_i; \theta)$ is the output of the current model. $T$ is the temperature parameter that smooths the probability distributions. In our experiments, we set $T = 2$ as it provides a well-established balance between softening the output distribution and maintaining stable gradients during distillation. LwF ensures that the model retains its predictive performance for the best-performing groups while adapting to the worst-performing groups. The distillation loss aligns the outputs for best-performing groups across stages, maintaining stability and reducing the risk of performance degradation.

**Elastic Weight Consolidation**: EWC introduces a regularization term to prevent significant changes to parameters critical for the best-performing groups:

$$\mathcal{L}_{\mathrm{CL}}(\theta) = \frac{1}{2}\sum_{j=1}^{|\theta|} F_j(\theta_j - \theta_j^*)^2, \tag{5}$$

where $\theta_j^*$ represents the $j$-th neural weight learned during the first stage, $\theta_j$ refers to the $j$-th weight currently being optimized, and $F_j$ corresponds to $j$-th entry in the diagonal of the Fisher information matrix (Jaynes, 2003), which quantifies the importance of the corresponding weight $\theta_j$. In the context of bias mitigation, we postulate that EWC helps to balance performance between the best-performing and the worst-performing groups by selectively penalizing changes to parameters critical for best-performing groups. By doing so, it minimizes the risk of degrading accuracy on advantaged groups while allowing updates to optimize for worst-performing groups. The Fisher information is computed empirically using samples from the best-performing groups ($\mathcal{G}_{\mathrm{best}}$). Following Van de Ven & Tolias (2019), it is estimated by averaging the squared gradients of the loss with respect to the model parameters, evaluated at the true labels. This serves as a measure of parameter sensitivity, ensuring that the model adapts in a way that retains prior knowledge.

In summary, BM-CL views fairness as an incremental constraint, where each adjustment to reduce bias between groups is treated as a new task, consistent with how training stages are defined in continual learning. The model thus aims to improve fairness over time while preserving previously learned capabilities. This formulation aligns with the stability–plasticity trade-off in continual learning (Wang et al., 2024), allowing the use of continual learning techniques to balance performance preservation with the progressive reduction of biases.

In BM-CL, this balance is governed by the regularization strength $\lambda$, which controls the degree to which the model may deviate from its behaviour of the first stage: large values enforce stability and preserve performance on $\mathcal{G}_{\mathrm{best}}$, whereas smaller values allow greater plasticity to improve on $\mathcal{G}_{\mathrm{worst}}$. A theoretical justification of this mechanism is provided in Appendix C, where we formally derive the bounded-harm guarantee induced by BM-CL and show how this limits leveling-down.

## 3 EXPERIMENTS

### 3.1 DATASETS

We evaluate BM-CL on both synthetic and real-world datasets, covering different data modalities:

- *Waterbirds* (Wah et al., 2011; Sagawa et al., 2019): consists of $11,788$ bird images labeled as *waterbird* and *landbird*, with a background attribute (*water* or *land*). The dataset is explicitly designed to simulate spurious correlations between label and background, which results in a significant performance disparity is groups where this association does not hold. We use the publicly available train, validation and test splits.
- *CelebA* (Liu et al., 2015): contains over $200,000$ celebrity face images annotated with $40$ binary attributes, such as hair color, eyeglasses, gender and facial expressions. We focus on predicting blond hair, where gender (male or female) introduces spurious correlations with the label. We also use the standard splits for training and evaluation.

| Dataset | Method | | Global Acc. | Balanced Acc. | Best Group | Worst Group | ↓ Disparity | ↓ LDE | ↑ IW |
|---|---|---|---|---|---|---|---|---|---|
| Waterbirds | Baseline | ERM | $88.2 \pm 0.5$ | $86.6 \pm 0.6$ | $99.5 \pm 0.1$ | $72.8 \pm 1.3$ | 26.7 | – | – |
| | BM | IRM | $91.2 \pm 2.1$ | $90.0 \pm 1.5$ | $98.7 \pm 1.4$ | $82.2 \pm 1.2$ | 16.4 | 0.9 | 9.4 |
| | | GroupDRO | $\mathbf{91.5} \pm 0.3$ | $\mathbf{90.2} \pm 0.2$ | $98.6 \pm 0.2$ | $82.6 \pm 0.4$ | 16.0 | 0.9 | 9.8 |
| | | ReSample | $90.5 \pm 0.9$ | $89.7 \pm 0.3$ | $94.9 \pm 1.1$ | $\mathbf{85.5} \pm 1.5$ | **9.5** | 4.6 | **12.6** |
| | | JTT | $88.8 \pm 0.6$ | $88.7 \pm 0.3$ | $96.2 \pm 0.5$ | $83.5 \pm 1.0$ | 12.8 | 3.3 | 10.7 |
| | BM-CL (ours) | GroupDRO-LwF | $90.0 \pm 0.6$ | $89.3 \pm 0.4$ | $99.0 \pm 0.2$ | $81.6 \pm 1.0$ | 17.4 | 0.5 | 8.8 |
| | | GroupDRO-EWC | $90.2 \pm 0.5$ | $89.4 \pm 0.4$ | $99.0 \pm 0.3$ | $81.2 \pm 1.2$ | 17.8 | 0.5 | 8.4 |
| | | ReSample-LwF | $89.6 \pm 0.6$ | $88.8 \pm 0.3$ | $99.3 \pm 0.2$ | $79.5 \pm 1.5$ | 19.8 | $\mathbf{0.2}^{\dagger}$ | 6.7 |
| | | ReSample-EWC | $90.9 \pm 0.4$ | $89.2 \pm 0.1$ | $99.3 \pm 0.2$ | $78.5 \pm 1.4$ | 20.7 | $\mathbf{0.2}^{\dagger}$ | 5.7 |
| CelebA | Baseline | ERM | $\mathbf{95.5} \pm 0.1$ | $82.1 \pm 0.6$ | $99.3 \pm 0.1$ | $46.1 \pm 2.2$ | 53.2 | – | – |
| | BM | IRM | $93.5 \pm 0.4$ | $88.3 \pm 0.7$ | $96.3 \pm 0.5$ | $71.1 \pm 2.1$ | 25.2 | 3.1 | 25.0 |
| | | GroupDRO | $93.5 \pm 0.4$ | $88.4 \pm 1.0$ | $95.8 \pm 0.3$ | $72.1 \pm 3.3$ | 23.7 | 3.5 | 26.0 |
| | | ReSample | $91.9 \pm 0.3$ | $89.4 \pm 0.6$ | $92.9 \pm 0.4$ | $80.1 \pm 2.4$ | 12.8 | 6.4 | 34.0 |
| | | JTT | $88.8 \pm 0.4$ | $86.8 \pm 0.6$ | $90.8 \pm 1.1$ | $76.7 \pm 0.7$ | 14.1 | 8.5 | 30.6 |
| | BM-CL (ours) | GroupDRO-LwF | $93.9 \pm 0.4$ | $88.7 \pm 0.6$ | $96.5 \pm 0.5$ | $72.4 \pm 2.4$ | 24.0 | **2.8** | 26.3 |
| | | GroupDRO-EWC | $93.7 \pm 0.2$ | $89.3 \pm 0.4$ | $95.9 \pm 0.4$ | $75.2 \pm 1.6$ | 20.6 | 3.5 | 29.1 |
| | | ReSample-LwF | $92.5 \pm 0.4$ | $90.0 \pm 0.2$ | $94.3 \pm 0.5$ | $80.8 \pm 1.2$ | 13.5 | 5.1 | 34.7 |
| | | ReSample-EWC | $92.1 \pm 0.3$ | $\mathbf{90.2} \pm 0.3$ | $93.6 \pm 0.3$ | $\mathbf{82.2} \pm 1.3$ | **11.4** | 5.7 | **36.1** |
| CheXpert | Baseline | ERM | $76.0 \pm 0.7$ | $75.5 \pm 0.7$ | $\mathbf{84.1} \pm 1.6$ | $67.1 \pm 2.0$ | 17.1 | – | – |
| | BM | IRM | $76.5 \pm 0.4$ | $76.5 \pm 0.3$ | $82.2 \pm 1.3$ | $72.0 \pm 1.4$ | 10.2 | 1.9 | 5.0 |
| | | GroupDRO | $76.4 \pm 0.4$ | $76.3 \pm 0.6$ | $81.3 \pm 0.9$ | $\mathbf{73.6} \pm 1.1^{\dagger}$ | **7.7** | 2.8 | $\mathbf{6.5}^{\dagger}$ |
| | | ReSample | $76.1 \pm 0.3$ | $76.2 \pm 0.1$ | $81.0 \pm 1.0$ | $72.7 \pm 1.3$ | 8.3 | 3.1 | 5.6 |
| | | JTT | $71.6 \pm 1.2$ | $71.6 \pm 1.0$ | $79.0 \pm 1.4$ | $67.3 \pm 1.1$ | 11.7 | 5.1 | 0.2 |
| | BM-CL (ours) | GroupDRO-LwF | $\mathbf{77.2} \pm 0.2^{\dagger}$ | $\mathbf{77.1} \pm 0.2^{\dagger}$ | $83.6 \pm 0.9$ | $73.0 \pm 1.7$ | 10.6 | 0.5 | 4.9 |
| | | GroupDRO-EWC | $76.2 \pm 0.6$ | $76.0 \pm 0.5$ | $81.8 \pm 0.7$ | $71.7 \pm 0.6$ | 10.2 | 2.3 | 4.6 |
| | | ReSample-LwF | $\mathbf{77.3} \pm 0.1^{\dagger}$ | $\mathbf{77.2} \pm 0.3^{\dagger}$ | $82.8 \pm 1.6$ | $\mathbf{73.5} \pm 2.0^{\dagger}$ | 9.3 | 1.3 | $\mathbf{6.4}^{\dagger}$ |
| | | ReSample-EWC | $76.0 \pm 0.5$ | $76.0 \pm 0.6$ | $81.5 \pm 1.4$ | $72.8 \pm 0.9$ | 8.8 | 2.6 | 5.7 |
| Adult | Baseline | ERM | $\mathbf{84.5} \pm 0.1$ | $71.1 \pm 0.4$ | $98.8 \pm 0.3$ | $32.5 \pm 2.8$ | 66.3 | – | – |
| | BM | IRM | $78.8 \pm 0.9$ | $82.3 \pm 1.1$ | $85.1 \pm 1.1$ | $87.5 \pm 4.4$ | 2.4 | 13.8 | 55.0 |
| | | GroupDRO | $79.6 \pm 0.6$ | $83.0 \pm 1.3$ | $87.7 \pm 1.0$ | $85.0 \pm 5.6$ | 2.7 | 11.2 | 52.5 |
| | | ReSample | $77.5 \pm 1.1$ | $82.1 \pm 1.5$ | $84.6 \pm 2.6$ | $86.2 \pm 5.2$ | 1.6 | 14.2 | 53.8 |
| | | JTT | $67.4 \pm 1.0$ | $74.7 \pm 1.0$ | $83.9 \pm 3.4$ | $\mathbf{93.8} \pm 4.4$ | 9.8 | 14.9 | **61.2** |
| | BM-CL (ours) | GroupDRO-LwF | $80.4 \pm 0.4$ | $83.7 \pm 0.6$ | $89.8 \pm 0.2$ | $86.2 \pm 5.2$ | 3.6 | $\mathbf{9.0}^{\dagger}$ | 53.8 |
| | | GroupDRO-EWC | $80.0 \pm 0.6$ | $\mathbf{84.3} \pm 0.7$ | $88.8 \pm 1.0$ | $90.0 \pm 5.6$ | 1.2 | 10.0 | 57.5 |
| | | ReSample-LwF | $79.9 \pm 0.7$ | $84.1 \pm 0.8$ | $89.8 \pm 0.3$ | $88.8 \pm 2.8$ | **1.0** | $\mathbf{9.1}^{\dagger}$ | 56.2 |
| | | ReSample-EWC | $79.5 \pm 0.4$ | $84.1 \pm 0.2$ | $87.8 \pm 0.8$ | $91.2 \pm 3.4$ | 3.5 | 11.1 | 58.8 |

Table 1: Comparison of BM-CL against the ERM baseline and state-of-the-art bias mitigation methods across datasets. The best result for each metric is highlighted in bold and the smallest degradation in best-group accuracy is highlighted in blue. LDE: Leveling-down Effect; IW: Improvement Worst. Results within 0.1 of the best are treated as comparable and marked with †. Best and worst groups correspond to those identified by ERM.

- *CheXpert* (Irvin et al., 2019): comprises over $224,316$ chest radiographs from $65,240$ patients annotated with 14 medical observations. We study binary classification of "Pleural effusion", considering patient age as the demographic attribute categorized into three groups: $< 40$, $40$–$65$, and $> 65$. We use only frontal images and randomly split the dataset into $70/10/20$ (training/validation/test), ensuring no patient overlap between splits.

- *Adult* (Becker & Kohavi, 1996): contains $48,842$ tabular instances with 14 categorical and numerical features, including education level, occupation, age and income. We perform a binary classification task predicting whether annual income of an individual exceeds \$50K, using a subset of 10 features. We construct intersectional demographic groups by combining race and sex. Following standard practice, we remove samples with missing values and also restrict the race attribute to *black* and *white*. After preprocessing, the dataset results in $30,940$ instances. This dataset was randomly divided into training, validation, and test sets using a $60/20/20$ ratio, stratified by the target label.

These datasets capture diverse sources of bias: Waterbirds and CelebA emphasize spurious correlations, while CheXpert and Adult highlights demographic imbalance. Dataset statistics are provided in Appendix A.

## 3.2 IMPLEMENTATION DETAILS

For all experiments on image data, we adopt a ResNet-50 architecture (He et al., 2016) pre-trained on ImageNet (Russakovsky et al., 2015) as the feature extractor. The final classification layer is

replaced with a fully connected layer with two output units to perform the binary classification task. For the tabular Adult dataset, we use a multilayer perceptron (MLP) composed of two fully connected layers with 256 units and ReLU activations. All experiments are implemented in PyTorch (Paszke et al., 2017) and executed on an NVIDIA Titan X GPU.[1].

We train all models to minimize the cross-entropy loss using stochastic gradient descent with momentum of 0.9 and weight decay of $10^{-4}$. We fix the number of training epochs to 30 for Waterbirds and 50 for CelebA, CheXpert and and Adult. The validation set is used for hyperparameter tuning and model selection, considering the best worst-group accuracy as the selection criterion. All hyperparameters are tuned via grid search, using predefined search ranges for the pretraining ratio $\rho$ and the regularization strength $\lambda$. We also apply early stopping with a patience of 10 epochs to prevent overfitting. The batch size is set to 32 for image datasets and 128 for tabular data. The learning rate is set to $10^{-3}$ for Waterbirds, CheXpert and Adult, and $10^{-4}$ for CelebA.

### 3.3 BASELINE MODELS AND PERFORMANCE EVALUATION

We evaluate variants of BM-CL with LwF and EWC against standard baselines, including the standard ERM trained until convergence and state-of-the-art bias mitigation methods (IRM, GroupDRO, ReSample and JTT). Each model is run 5 times with different random seeds to account for variability.

Performance is measured using: i) global accuracy (the overall accuracy on the full test set), ii) balanced accuracy (the mean of group-wise accuracies), iii) accuracies of the best- and worst-performing groups, and iv) their disparity. We also report the improvements in worst-group accuracy and the degradation in best-group accuracy relative to ERM. Best and worst groups are identified from ERM results and kept fixed across comparisons to assess the effect of bias mitigation strategies on the initially advantaged or disadvantaged groups.

## 4 RESULTS

### 4.1 COMPARISON WITH BASELINES AND STATE-OF-THE-ART METHODS

Table 1 compares BM-CL with ERM and state-of-the-art bias mitigation methods on Waterbirds, CelebA, CheXpert and Adult across 5 runs. We report global accuracy and balanced accuracy, best- and worst-performing groups, disparity and two relative measures: the leveling-down effect (degradation in best-group accuracy relative to ERM) and the worst-group improvement (gain over ERM). Overall, BM-CL , particularly with LwF regularization, consistently achieves the lowest leveling-down effect (highlighted in blue) while delivering competitive gains in worst-group performance.

In Waterbirds, GroupDRO achieves the highest global (91.5%) and balanced (90.2%) accuracies, while ReSample achieves the highest worst-group accuracy (85.5%). Among the bias mitigation methods, IRM achieves the highest best-group accuracy (98.7%). BM-CL methods perform competitively: ReSample-LwF and ReSample-EWC nearly match ERM in best-group accuracy (99.3% vs. 99.5%), while GroupDRO-LwF improves worst-group accuracy to 81.6% with minimal degradation in the best group (99.0%).

In CelebA, ERM obtains the highest global accuracy (95.5%) and best-group accuracy (99.3%), but suffers from a large performance gap, with worst-group accuracy dropping to 46.1%. GroupDRO, ReSample, and JTT reduce this disparity, but at the cost of best-group accuracy. In contrast, IRM exhibits smaller leveling-down, but offering limited improvement for the disadvantaged groups. BM-CL balances both: GroupDRO-LwF preserves best-group accuracy (96.5%) with the smallest leveling-down effect, while ReSample-EWC achieves the highest worst-group accuracy (82.2%) and better preserves best-group performance than ReSample alone.

In CheXpert, ERM again exhibits a large disparity between best- and worst-performing groups (84.1% vs. 67.1%). GroupDRO and ReSample improve worst-group accuracy (up to 73.6% and 72.7%), but BM-CL provides the most balanced trade-offs. GroupDRO-LwF and ReSample-LwF achieve the highest global/balanced accuracies and strong worst-group results (73.0%-73.5%), with

---

[1]Code is publicly available at `https://anonymous.4open.science/r/BM-CL`

minimal degradation in best-group accuracy (83.6%). IRM again provides strong best-group performance between the bias-mitigation baselines, but it still underperforms BM-CL.

In the Adult dataset, bias mitigation methods clearly improve the poor worst-group performance of ERM, with JTT achieving the highest accuracy (93.8%). BM-CL methods also outperform both GroupDRO and ReSample in the worst group, with the EWC-based variants exceeding 90% accuracy. Regarding leveling-down, BM-CL exhibits the smallest reduction in best-group performance, particularly for the LwF-based variants (89.8% accuracy), which preserve accuracy more effectively than conventional bias mitigation approaches.

In summary, while bias mitigation methods improve disadvantaged groups at the expense of advantaged ones, BM-CL consistently limits the leveling-down effect, achieving fairer and more reliable performance across groups. Additional results comparing best- and worst-performing groups by run and subgroup-level performance are presented in Appendix B.

### 4.2 Ablation study: impact of pretaining ratio and CL regularization

We analyze the effect of the two main hyperparameters in BM-CL: the pretraining ratio ($\rho$) and the CL regularization strength ($\lambda$). Specifically, we test $\rho$ in $\{0.1, 0.2, 0.3\}$ and $\lambda$ in $\{0.0, 0.1, 1.0, 10.0\}$. In the case of EWC, since it penalizes deviations in weights directly, its values are scaled by $10^3$ to ensure comparable effects.

Fig. 3 shows the mean validation accuracy on Waterbirds over 3 runs, using GroupDRO for bias mitigation and LwF for CL. Our results highlight the trade-off of $\lambda$. Strong regularization helps maintain high performance for the best-performing groups, mitigating the degradation that typically follows in stage 2. In contrast, weaker regularization tend to benefit the worst-performing groups, likely because weaker regularization allows more flexibility during bias mitigation. The pretraining ratio $\rho$ shows little overall impact, except under strong regularization ($\lambda = 10$), where increasing $\rho$ from 0.1 to 0.3 improves worst-group accuracy from 72.7% to 77.5%. This suggests that when flexibility during fine-tuning is limited, pretraining becomes more critical.

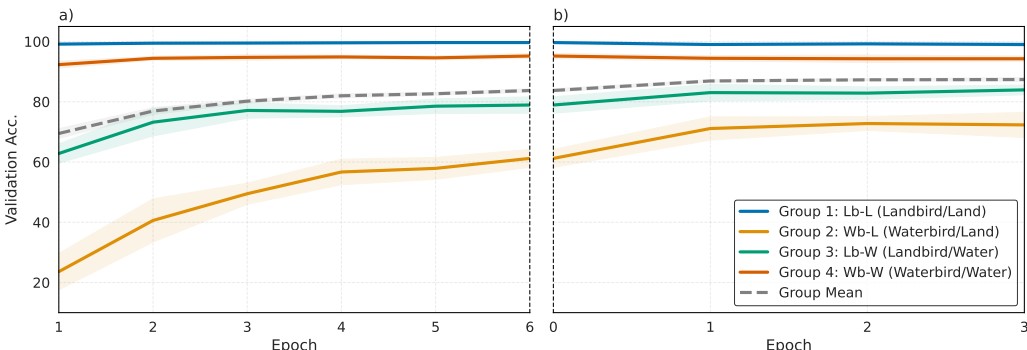

Figure 2: Accuracy (mean $\pm$ standard deviation) in the validation set during training with BM-CL on the Waterbirds dataset. a) Initial training by standard ERM, b) Fine-tuning using BM-CL.

To better understand how BM-CL supports bias mitigation while reducing catastrophic forgetting, Fig. 2 tracks validation accuracy during training. In stage 1 (ERM), the model quickly fits majority-correlated groups (Lb-L, Wb-W), leaving mismatched groups (Wb-L, Lb-W) underperforming. In stage 2 (BM-CL fine-tuning), the CL regularization retains advantaged-groups performance while the bias mitigation method (GroupDRO, in this case) guides the model to improve performance on the worst-performing groups. This manifests as increasing validation accuracy on the worst-performing groups during the course of fine-tuning.

## 5 Conclusion

This study introduced Bias Mitigation through Continual Learning (BM-CL), a novel framework that integrates continual learning with bias mitigation strategies to address the prevalent leveling-

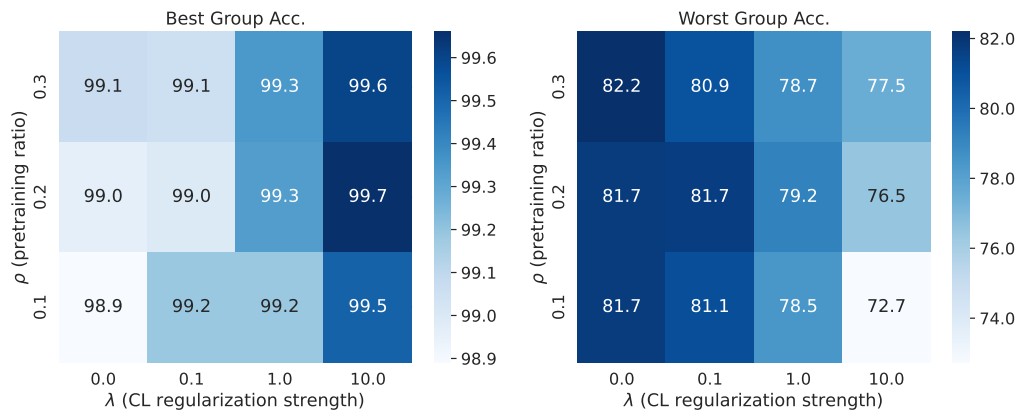

Figure 3: Mean accuracy of BM-CL in the validation set on the Waterbirds dataset comparing the pretraining ratio ($\rho$) and CL regularization strength ($\lambda$) across the best- and the wort-performing groups.

down effect in machine learning fairness interventions. Our key insight was to reinterpret bias mitigation as a form of task-incremental continual learning, allowing models to improve outcomes for disadvantaged groups while preserving performance for advantaged groups. By incorporating Learning without Forgetting (LwF) and Elastic Weight Consolidation (EWC), BM-CL mitigates the risk of catastrophic forgetting that occurs when fairness objectives shift model priorities.

Our experiments on both synthetic and real-world datasets, including Waterbirds, CelebA, CheXpert and Adult, demonstrated that BM-CL consistently improves worst group accuracy while minimizing the performance trade-offs for best group accuracy typically observed in conventional bias mitigation techniques. Notably, LwF-augmented methods preserved best-group accuracy to a greater extent than other approaches, effectively balancing fairness and overall accuracy. Our findings suggest that continual learning principles provide a promising mechanism for developing fairer machine learning models without exacerbating accuracy degradation in advantaged groups. By framing bias mitigation as a continual learning problem, this study opens new pathways for leveraging the extensive toolkit of continual learning techniques to improve fairness without compromising model reliability.

Finally, BM-CL provides a foundation for several promising research directions. Extending the framework to settings without explicit group labels, to richer or overlapping demographic attributes may broaden its applicability. Another direction is to explore additional CL methods, such as replay-based approaches, parameter-isolation techniques or meta-learning strategies, to further control the stability-plasticity trade-off in fairness settings. Moreover, integrating the principles of CL with alternative fairness notions, such as calibration or individual fairness, represents an interesting opportunity to extend BM-CL beyond group-fairness.

## REPRODUCIBILITY STATEMENT

The code to reproduce all experiments in this work has been submitted to an anonymous repository to ensure reproducibility during the review process. Instructions for running the experiments are provided in the README. All datasets used are publicly available, and we include detailed instructions for downloading and preprocessing them in the repository.

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

## A    DATASET DETAILS

Table 2 provides a detailed summary of key statistics for each dataset, including the number of samples in the training, validation, and test sets, as well as the sizes of the largest and smallest groups.

| Dataset | Training | Max group | Min group | Classes | Attrs |
|---|---|---|---|---|---|
| Waterbirds | 4,795 | 3,498 | 56 | 2 | 2 |
| CelebA | 162,770 | 71,629 | 1,387 | 2 | 2 |
| CheXpert | 133,785 | 35,502 | 5,521 | 2 | 3 |
| Adult | 18,564 | 7,807 | 59 | 2 | 4 |

Table 2: Summary of datasets used in our experiments. The maximum and minimum group sizes refer to the training set.

## B    ADDITIONAL RESULTS

Fig. 4 provides a comparative view of best- and worst-group accuracy across multiple runs and datasets. Each point in the scatter plot represents a single run, with the x-axis showing the accuracy on the best-performing group and the y-axis showing the accuracy on the worst-performing group. We use color to distinguish methods that incorporate continual learning (CL) from those that do not, and include dashed lines to indicate the mean accuracy along each axis for visual reference.

The figures show the trade-off between mitigating bias and avoiding the leveling-down effect. Traditional methods often succeed in boosting underperforming groups but struggle to preserve accuracy on groups that already perform well. In contrast, BM-CL variants achieve a more favorable trade-off. This is evidenced by the fact that points corresponding to CL methods mostly stay at the upper-right part of the plot, indicating that they not only mitigate bias effectively but also minimize the leveling-down effect. Note that in Adult the worst-group accuracy exhibits more discrete fluctuations across seeds. This is expected because the worst-performing subgroup is also the smallest one (15 samples in the test set), making its accuracy more sensitive to individual prediction changes.

Tables 3, 4, 5 and 6 detail subgroup-level accuracies for each dataset, providing a more granular view of how each method performs across different demographic or contextual combinations in terms of fairness and performance preservation. For Waterbirds, both ReSample-LwF and GroupDRO-LwF improve accuracy in the mismatched subgroups (Wb-L and Lb-W), with minimal leveling-down on the dominant group (Lb-L). In the case of CelebA, the largest disparity appears between blond males (Bh-M) and non-blond males (Nh-M). BM-CL boosts Bh-M accuracy from 46.1% (ERM) to 80.8% (ReSample-LwF) and 82.2% (ReSample-EWC), while keeping high accuracy in Nh-M (94.3% and 93.6%, respectively).

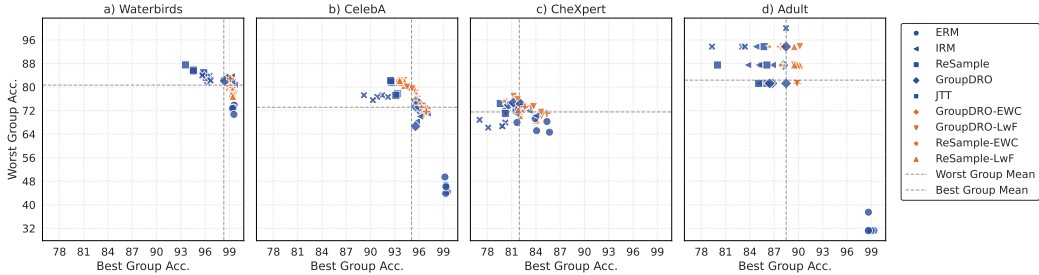

Figure 4: Comparison of the accuracy for each run and for best and worst groups across datasets. ERM and traditional bias mitigation methods (IRM, GroupDRO, ReSample, and JTT, in blue) are contrasted with BM-CL (GroupDRO-LwF, ReSample-LwF, GroupDRO-EWC, and ReSample-EWC, in orange).

In CheXpert, performance is more distributed across subgroups. BM-CL leads to significant gains in underperforming cases like young patients with pleural effusion (Pe-Y), reaching 72.4% (GroupDRO-LwF) and 73.8% (ReSample-LwF), compared to ERM (67.5%). Accuracy in the best-performing group, young patients without pleural effusion (Ne-Y), remains high for BM-CL, with minimal leveling-down. GroupDRO-EWC and ReSample-EWC also improve Pe-Y performance (71.6% and 72.0%, respectively), supporting the utility of BM-CL in complex medical datasets.

Finally, in the Adult dataset, BM-CL also shows clear improvements on the most underrepresented subgroups, alleviating the severe underperformance exhibited by ERM in groups such as Hi-BF (black females with high income). Among the BM-CL methods, the EWC variants achieve the strongest gains: GroupDRO-EWC reaches 90.0%, and ReSample-EWC attains 91.2% in this subgroup. At the same time, BM-CL better preserves performance in the best-performing group Li-BF (black females with low income), achieving lower leveling-down than standard GroupDRO and ReSample. In particular, both LwF variants maintain the highest best-group accuracy (89.8%) while still improving worst-group performance. These results show that BM-CL provides a more favorable trade-off on tabular data as well.

| | Method | Lb-L | Wb-L | Lb-W | Wb-W |
|---|---|---|---|---|---|
| Baseline | ERM | $99.5 \pm 0.1$ | $72.8 \pm 1.3$ | $79.6 \pm 1.0$ | $94.5 \pm 0.3$ |
| BM | IRM | $98.7 \pm 1.4$ | $82.2 \pm 1.2$ | $85.4 \pm 3.7$ | $93.9 \pm 0.8$ |
| | GroupDRO | $98.6 \pm 0.2$ | $82.6 \pm 0.4$ | $86.3 \pm 0.8$ | $93.2 \pm 0.5$ |
| | ReSample | $94.9 \pm 1.1$ | $85.5 \pm 1.5$ | $87.3 \pm 1.9$ | $91.1 \pm 0.4$ |
| | JTT | $96.2 \pm 0.5$ | $83.5 \pm 1.0$ | $81.6 \pm 1.6$ | $93.3 \pm 0.5$ |
| BM-CL (ours) | GroupDRO-LwF | $99.0 \pm 0.2$ | $81.6 \pm 1.0$ | $82.2 \pm 1.4$ | $94.5 \pm 0.4$ |
| | GroupDRO-EWC | $99.0 \pm 0.3$ | $81.2 \pm 1.2$ | $82.6 \pm 1.3$ | $94.6 \pm 0.5$ |
| | ReSample-LwF | $99.3 \pm 0.2$ | $79.5 \pm 1.5$ | $81.4 \pm 1.7$ | $94.9 \pm 0.5$ |
| | ReSample-EWC | $99.3 \pm 0.2$ | $78.5 \pm 1.4$ | $85.3 \pm 1.5$ | $93.8 \pm 0.7$ |

Table 3: Subgroup-level accuracy (mean $\pm$ standard deviation) on the Waterbirds dataset. Subgroups are defined by bird type and background: landbird on land (Lb-L), waterbird on land (Wb-L), landbird on water (Lb-W) and waterbird on water (Wb-W).

| | Method | Nh-F | Bh-F | Nh-M | Bh-M |
|---|---|---|---|---|---|
| Baseline | ERM | $95.7 \pm 0.3$ | $87.2 \pm 0.9$ | $99.3 \pm 0.1$ | $46.1 \pm 2.2$ |
| BM | IRM | $91.7 \pm 0.4$ | $94.0 \pm 0.6$ | $96.3 \pm 0.5$ | $71.1 \pm 2.1$ |
| | GroupDRO | $92.0 \pm 0.7$ | $93.7 \pm 0.5$ | $95.8 \pm 0.3$ | $72.1 \pm 3.3$ |
| | ReSample | $90.9 \pm 0.5$ | $93.7 \pm 0.9$ | $92.9 \pm 0.4$ | $80.1 \pm 2.4$ |
| | JTT | $86.4 \pm 0.5$ | $93.4 \pm 1.8$ | $90.8 \pm 1.1$ | $76.7 \pm 0.7$ |
| BM-CL (ours) | GroupDRO-LwF | $92.3 \pm 0.8$ | $93.7 \pm 1.1$ | $96.5 \pm 0.5$ | $72.4 \pm 2.4$ |
| | GroupDRO-EWC | $92.5 \pm 0.5$ | $93.7 \pm 0.6$ | $95.9 \pm 0.4$ | $75.2 \pm 1.6$ |
| | ReSample-LwF | $91.0 \pm 0.7$ | $94.2 \pm 0.9$ | $94.3 \pm 0.5$ | $80.8 \pm 1.2$ |
| | ReSample-EWC | $90.4 \pm 0.6$ | $94.8 \pm 0.4$ | $93.6 \pm 0.3$ | $82.2 \pm 1.3$ |

Table 4: Subgroup-level accuracy (mean $\pm$ standard deviation) on the CelebA dataset. Subgroups are defined by hair color and gender: Not blond female (Nh-F), blond female (Bh-F), not blond male (Nh-M), and blond male (Bh-M).

| | Method | Ne-Y | Pe-Y | Ne-M | Pe-M | Ne-O | Pe-O |
|---|---|---|---|---|---|---|---|
| Baseline | ERM | $84.1 \pm 1.6$ | $67.5 \pm 2.4$ | $78.2 \pm 2.4$ | $75.2 \pm 2.3$ | $70.9 \pm 2.4$ | $77.3 \pm 2.4$ |
| BM | IRM | $82.2 \pm 1.3$ | $72.5 \pm 1.4$ | $77.5 \pm 1.0$ | $76.5 \pm 1.6$ | $73.2 \pm 1.3$ | $76.9 \pm 1.3$ |
| | GroupDRO | $81.3 \pm 0.9$ | $73.0 \pm 1.4$ | $77.5 \pm 0.6$ | $76.7 \pm 1.4$ | $74.0 \pm 0.9$ | $75.6 \pm 1.7$ |
| | ReSample | $81.0 \pm 1.0$ | $73.6 \pm 1.3$ | $77.3 \pm 1.5$ | $75.9 \pm 1.4$ | $73.5 \pm 1.7$ | $75.7 \pm 1.8$ |
| | JTT | $79.0 \pm 1.4$ | $67.2 \pm 1.2$ | $73.6 \pm 1.6$ | $71.1 \pm 1.6$ | $67.5 \pm 1.8$ | $71.4 \pm 1.7$ |
| BM-CL (ours) | GroupDRO-LwF | $83.6 \pm 0.9$ | $72.4 \pm 1.7$ | $78.4 \pm 0.4$ | $77.5 \pm 0.3$ | $73.7 \pm 1.4$ | $77.2 \pm 1.2$ |
| | GroupDRO-EWC | $81.8 \pm 0.7$ | $71.6 \pm 0.6$ | $77.6 \pm 1.3$ | $75.9 \pm 0.9$ | $73.5 \pm 1.4$ | $75.7 \pm 1.8$ |
| | ReSample-LwF | $82.8 \pm 1.6$ | $73.8 \pm 2.1$ | $79.0 \pm 1.1$ | $77.0 \pm 1.4$ | $74.3 \pm 0.8$ | $76.5 \pm 1.2$ |
| | ReSample-EWC | $81.5 \pm 1.4$ | $72.0 \pm 2.1$ | $76.6 \pm 0.5$ | $77.0 \pm 1.1$ | $71.3 \pm 1.0$ | $77.8 \pm 1.2$ |

Table 5: Subgroup-level accuracy (mean $\pm$ standard deviation) on the CheXpert dataset. Subgroups are defined by age group (Young, Middle, Old) and presence of pleural effusion (Pe) or absence (Ne).

| | Method | Li-BF | Hi-BF | Li-BM | Hi-BM | Li-WF | Hi-WF | Li-WM | Hi-WM |
|---|---|---|---|---|---|---|---|---|---|
| Baseline | ERM | $98.8 \pm 0.3$ | $32.5 \pm 2.8$ | $95.0 \pm 0.6$ | $50.8 \pm 2.6$ | $97.8 \pm 0.1$ | $42.0 \pm 1.0$ | $88.8 \pm 0.9$ | $62.8 \pm 2.5$ |
| BM | IRM | $85.1 \pm 1.1$ | $87.5 \pm 4.4$ | $86.6 \pm 1.2$ | $76.2 \pm 2.9$ | $82.8 \pm 1.6$ | $86.1 \pm 2.7$ | $73.6 \pm 2.0$ | $80.9 \pm 2.6$ |
| | GroupDRO | $87.7 \pm 1.0$ | $85.0 \pm 5.6$ | $88.0 \pm 1.0$ | $76.9 \pm 2.4$ | $84.7 \pm 0.8$ | $85.8 \pm 2.6$ | $73.6 \pm 0.7$ | $81.9 \pm 0.4$ |
| | ReSample | $84.6 \pm 2.6$ | $86.2 \pm 5.2$ | $85.7 \pm 1.0$ | $78.1 \pm 4.2$ | $82.0 \pm 2.9$ | $85.5 \pm 1.4$ | $69.1 \pm 1.0$ | $85.4 \pm 1.4$ |
| | JTT | $83.9 \pm 3.4$ | $93.8 \pm 4.4$ | $67.1 \pm 1.5$ | $71.9 \pm 5.4$ | $81.5 \pm 2.6$ | $78.9 \pm 2.7$ | $59.7 \pm 1.9$ | $61.1 \pm 2.4$ |
| BM-CL (ours) | GroupDRO-LwF | $89.8 \pm 0.2$ | $86.2 \pm 5.2$ | $87.3 \pm 0.6$ | $77.7 \pm 2.2$ | $86.6 \pm 0.5$ | $86.0 \pm 1.7$ | $74.0 \pm 1.4$ | $82.2 \pm 1.3$ |
| | GroupDRO-EWC | $88.8 \pm 1.0$ | $90.0 \pm 5.6$ | $86.7 \pm 1.6$ | $80.8 \pm 3.0$ | $85.6 \pm 1.0$ | $86.6 \pm 2.0$ | $73.5 \pm 1.6$ | $82.7 \pm 1.2$ |
| | ReSample-LwF | $89.8 \pm 0.3$ | $88.8 \pm 2.8$ | $86.1 \pm 2.4$ | $81.5 \pm 4.4$ | $87.2 \pm 0.6$ | $83.7 \pm 1.2$ | $71.9 \pm 1.7$ | $84.2 \pm 1.2$ |
| | ReSample-EWC | $87.8 \pm 0.8$ | $91.2 \pm 3.4$ | $86.5 \pm 1.5$ | $81.5 \pm 2.9$ | $85.4 \pm 0.6$ | $84.6 \pm 0.7$ | $72.2 \pm 1.1$ | $83.7 \pm 0.8$ |

Table 6: Subgroup-level accuracy (mean $\pm$ standard deviation) on the Adult dataset. Subgroups are defined by the intersection of race (Black, White), sex (Male, Female) and income level (Hi: high income, Li: low income).

## C  A LAGRANGIAN VIEW OF FAIRNESS–FORGETTING TRADE-OFFS

BM-CL addresses a core challenge in fairness: improving performance for disadvantaged groups without harming advantaged groups. Here, we propose to interpret the initial formulation we introduced in Eq. 2 as a Lagrangian relaxation of a constrained optimization problem that explicitly limits forgetting on initially advantaged groups:

$$\min_{\theta} \mathcal{L}_{\mathrm{BM}}(\theta) \quad \text{subject to} \quad \mathcal{L}_{\mathrm{CL}}(\theta) \leq \epsilon, \tag{6}$$

where $\epsilon \geq 0$ bounds the permissible performance degradation on advantaged groups.

The corresponding Lagrangian formulation for a fixed $\lambda \geq 0$ is then

$$\hat{\theta} = \arg \min_{\theta} \left[ \mathcal{L}_{\mathrm{BM}}(\theta) + \lambda \mathcal{L}_{\mathrm{CL}}(\theta) \right]. \tag{7}$$

Let $\theta^*$ be the ERM solution from stage 1. By optimality, we have

$$\mathcal{L}_{\mathrm{BM}}(\hat{\theta}) + \lambda \mathcal{L}_{\mathrm{CL}}(\hat{\theta}) \leq \mathcal{L}_{\mathrm{BM}}(\theta^*) + \lambda \mathcal{L}_{\mathrm{CL}}(\theta^*). \tag{8}$$

For both EWC and LwF, the continual learning loss satisfies $\mathcal{L}_{\mathrm{CL}}(\theta^*) = 0$:

- In **EWC**: $\mathcal{L}_{\mathrm{CL}}(\theta) = \frac{1}{2} \sum_{j=1}^{|\theta|} F_j (\theta_j - \theta_j^*)^2$, so $\mathcal{L}_{\mathrm{CL}}(\theta^*) = 0$.
- In **LwF**: $\mathcal{L}_{\mathrm{CL}}(\theta) = \mathrm{KL}(q^* \| q)$, where $q^*$ are reference predictions from $\theta^*$, so at $\theta = \theta^*$, $q = q^*$ and $\mathcal{L}_{\mathrm{CL}}(\theta^*) = 0$.

Using this property and rearranging, we obtain the trade-off bound

$$\mathcal{L}_{\mathrm{CL}}(\hat{\theta}) \leq \frac{\mathcal{L}_{\mathrm{BM}}(\theta^*) - \mathcal{L}_{\mathrm{BM}}(\hat{\theta})}{\lambda}. \tag{9}$$

This reveals the trade-off enforced by BM-CL: the potential harm to advantaged groups is bounded by the improvement achieved for disadvantaged groups, divided by $\lambda$. In other words, $\lambda$ controls the trade-off rate:

- **Small $\lambda$**: Favors bias mitigation, allowing larger $\mathcal{L}_{\mathrm{CL}}$ (more forgetting).
- **Large $\lambda$**: Favors knowledge preservation, constraining $\mathcal{L}_{\mathrm{CL}}$ but limiting bias improvement.

As we can see, any deviation from the ERM solution on advantaged groups is upper-bounded by the fairness improvement, quantified by $\mathcal{L}_{\mathrm{BM}}(\theta^*) - \mathcal{L}_{\mathrm{BM}}(\hat{\theta})$, scaled by $1/\lambda$. In other words, $\mathcal{L}_{\mathrm{BM}}(\theta^*) - \mathcal{L}_{\mathrm{BM}}(\hat{\theta})$ captures the gain in fairness, while $\mathcal{L}_{\mathrm{CL}}$ measures the permitted amount of forgetting enforced by the trade-off parameter $\lambda$.

Thus, the CL term explicitly bounds how much the model may deteriorate on previously best-performing groups as fairness is improved, providing a bounded-harm guarantee that helps prevent leveling-down.

