# OpenReview forum: "BM-CL: Bias Mitigation through the lens of Continual Learning"
_ICLR.cc/2026/Conference — Submitted to ICLR 2026_

### Official Review · Reviewer_2D32 · 2025-10-28

**Soundness:** 3
**Presentation:** 2
**Contribution:** 2
**Rating:** 4
**Confidence:** 3

**Summary:**

The paper advocates a novel BM-CL framework that reinterprets bias mitigation as a continual learning (CL) problem. The authors draw parallels between the leveling-down effect in fairness interventions (where improving disadvantaged groups harms advantaged ones) and catastrophic forgetting in CL. The proposed BM-CL combines standard bias mitigation methods (GroupDRO, ReSample) with CL techniques (LwF and EWC). Experiments on Waterbirds, CelebA, and CheXpert datasets show improvements in worst-group accuracy with minimal degradation in advantaged groups.

**Strengths:**

+ The core idea of framing bias mitigation as continual learning is conceptually elegant. Drawing a formal analogy between the leveling-down effect and catastrophic forgetting provides a new view that could inspire cross-fertilization between fairness and lifelong learning research.

+ BM-CL integrates easily with existing bias-mitigation pipelines by adding a CL-style regularization term. Hence the approach is clear and reproducible, with minimal additional complexity for understanding or implementation.

+ Evaluations on multiple benchmark datasets (spurious correlation, demographic bias, medical imaging) are comprehensive and show consistent trends. The code availability and ablation studies increase reproducibility.

**Weaknesses:**

- While providing a new lens for understanding fairness-aware ML, the proposed method is largely a straightforward combination of existing techniques (LwF, EWC, and bias-mitigation baselines) with limited algorithmic innovation. The design omits potential drawbacks (e.g., stability-plasticity conflicts under multi-attribute settings, or extension to non-image modalities).

- No new theoretical framework or analytical insight beyond the analogy to forgetting is developed. There also lacks justification for why continual-learning regularizers specifically address the fairness-accuracy trade-off beyond empirical correlation.

- While the results show improvements, the absolute differences over strong baselines (GroupDRO, ReSample) are often small (~1-2% balanced accuracy). Quantitative gains are marginal.

- It is unclear whether these improvements are statistically significant or robust across seeds and datasets.

**Questions:**

1. How does BM-CL perform when the number of demographic groups increases or when group definitions overlap (intersectional fairness)?

2. Can compare BM-CL against more recent fairness methods (e.g., invariant risk minimization, domain generalization-based fairness) to contextualize competitiveness?

3. Could the authors provide a theoretical or empirical justification for why the continual-learning regularization term directly mitigates "leveling-down" beyond preventing weight drift?

4. How sensitive are results to the choice of hyperparameters such as λ (regularization strength) and pretraining ratio ρ across datasets? Do these require dataset-specific tuning?

5. Have examined the effect of CL regularization on model calibration or other fairness metrics (equalized odds, demographic parity, delta EO or DP)?

**Details Of Ethics Concerns:**

The ethical framing ("positive-sum fairness") is interesting but not substantiated with theoretical or societal analysis.

---

> ### Author Response · Authors · 2025-11-21
> **1/2**
>
> We thank the reviewer for their feedback and highlighting that our work “is a novel framework”, “is conceptually relevant”, that it “integrates easily with existing bias-mitigation strategies”, and that it is “clear and reproducible, with minimal additional complexity”.
>
> **W1. Algorithmic innovation**
>
> We emphasize that our contribution is not the introduction of yet another bias mitigation algorithm, but a new formulation that casts fairness as a continual learning problem. This connection between fairness and continual learning has not been explored in prior work and enables analyzing fairness degradation through the lens of the stability–plasticity trade-off. BM-CL is a general framework built around this perspective, rather than a single-purpose algorithm. By reframing fairness degradation as a form of catastrophic forgetting, we establish a formal link between two previously disconnected research areas and open the door for cross-pollination between the fairness and continual learning communities.
>
> Although EWC and LwF are well-established continual learning tools, their reinterpretation as mechanisms that explicitly constrain forgetting on advantaged groups is new. Importantly, we not only propose this conceptual framework, but also demonstrate its practical value: BM-CL effectively modulates the worst–best group trade-off across three datasets, outperforming or matching prior bias mitigation approaches. Overall, BM-CL is a flexible and modular framework (agnostic to the specific mitigation method) and this generality is a core part of the contribution.
>
> **W2 and Q3. Theoretical justification**
>
> We agree with the reviewer in that our paper was lacking a more in-depth theoretical justification. That is why we have now added a new section (see the new Appendix C in the revised manuscript) where we propose a Lagrangian view of the fairness-forgetting trade-off, by interpreting the initial formulation we introduced in Eq.2  as a Lagrangian relaxation of a constrained optimization problem that explicitly limits forgetting on initially advantaged groups.This new view allows us to derive a formal bound on how much the model may deteriorate on previously best-performing groups as fairness is improved.
>
> Importantly, note that EWC penalizes changes to parameters important for advantaged groups via the Fisher information. LwF aligns stage-1 and stage-2 predictions through the KL divergence. This links fairness improvement to model stability, since the model adapts using parameters that are less critical for $G_{best}$. As shown in Section 4.2, setting $\lambda = 0$ removes the constraint and reproduces the leveling-down effect, supporting the stability-plasticity interpretation.
>
> **W3 and W4. Quantitative gain and robustness**
>
> Although improvements in the worst-group can appear small, BM-CL achieves these gains with minimal degradation performance in the best-group, unlike the baselines, which often trade off worst-group gains for harm to advantaged groups. Preventing such degradation is the practical benefit that BM-CL introduces, and this trend is consistently observed across datasets and baselines. We also note that leveling-down generates ethical dilemmas in many applications, such as in the medical domain. This means even small gains (by broader ML standards) can resolve ethical conflicts. For this reason the fairness community continues to work on bias mitigation tools that avoid leveling-down.
>
> Regarding statistical robustness, our reported results are averaged over five random seeds. While this is not sufficient to compute significance tests, the overall pattern (improved worst-group accuracy combined with reduced leveling-down) remains stable and consistent.

---

> > ### Author Response · Authors · 2025-11-21
> > **2/2**
> >
> > **Q1. Dependence on the number of demographic groups**
> >
> > In our formulation, the number of groups does not directly affect the optimization process of BM-CL. Groups influence only through its assignment to $G_{best}$ or $G_{worst}$, which determines where the CL regularizer is applied. The cost of both EWC and LwF is proportional to the number of samples in $G_{best}$. On the other hand, the identification step of best and worst groups scales linearly with the number of groups. Therefore, BM-CL does not suffer from scalability issues with respect to the number of groups beyond the common issues faced by all fairness approaches. What matters is the size and representativeness of the samples within $G_{best}$.
> >
> > **Q2. Comparison with more recent methods**
> >
> > We have added Invariant Risk Minimization (IRM) (Arjovsky et al., 2019) to the comparison in the revised manuscript. Results show that while IRM achieves lowest leveling-down compared to other bias mitigation methods, it still underperforms BM-CL.
> >
> > **Q4. Sensitivity of the regularization strength $\lambda$ and pretraining ratio $\rho$**
> >
> > The regularization strength $\lambda$ controls stability, moderate values (0.1, 1.0) will balance gains in fairness with preservation of prior accuracy for $G_{best}$. The pretraining ratio $\rho$ determines how close stage 1 is to the full ERM convergence, a small value (0.1-0.3) is enough to identify $G_{best}$ and $G_{worst}$ without requiring a long initial phase. Both hyperparameters have clear interpretations and broad stable ranges. In our experiments, we adopt dataset-specific tuning for fair comparison across baselines.
> >
> > **Q5. Calibration and fairness metrics**
> >
> > We follow the standard setting in bias-mitigation literature, where groups are defined by the intersection of attributes $\mathcal{A}$ and labels ($\mathcal{Y}$). In this setting, each subgroup corresponds to a single label, so differences in group accuracy directly captures fairness disparities. We acknowledge that extending the BM-CL analysis to calibration or other additional fairness metrics is a promising direction for future work, so we have added this idea in the future works section.
> >
> > **Main changes in the revised manuscript:** please note that we have uploaded a revised manuscript introducing changes based on your comments as well as the comments of the other reviewers. The new content is highlighted in violet. As a result, we have now included 1) a new benchmark method for bias mitigation (Invariant Risk Minimization), 2) a new section where we provide a more rigorous theoretical interpretation of our framework through a Lagrangian view of the fairness-forgetting trade off and 3) experiments on a new dataset of tabular data, to show that our framework can be applied to data modalities beyond images. We believe this revised manuscript is a much stronger version of our work. Should you agree that the revision improves the paper, we would appreciate your consideration in updating the score.

---

### Official Review · Reviewer_23mH · 2025-10-30

**Soundness:** 1
**Presentation:** 2
**Contribution:** 1
**Rating:** 2
**Confidence:** 3

**Summary:**

The paper proposes BM-CL, framing bias mitigation's "leveling-down" effect as a Continual Learning "forgetting" problem. It uses a two-stage approach: ERM training followed by fine-tuning with a bias mitigation loss and a CL regularizer (LwF/EWC) to preserve performance on initially advantaged groups.

**Strengths:**

The paper addresses a critical and practical problem in fairness. The "leveling-down" effect is a major barrier to deploying fair models in high-stakes settings where harming any group is unacceptable.

**Weaknesses:**

1.Limited Novelty / Incremental Contribution: The core idea of linking "leveling-down" to "forgetting" is an interesting perspective, but the proposed solution (BM-CL) primarily combines existing, off-the-shelf techniques: standard bias mitigation losses (GroupDRO, ReSample) with standard CL regularizers (LwF, EWC) in a sequential manner. There is little fundamental algorithmic innovation presented. The contribution feels incremental rather than introducing a truly new mechanism for bias mitigation.

2.Marginal Empirical Improvement: Even when compared against the selected baselines (which, as noted previously, might not be the most appropriate ones for evaluating FWH), the performance gains offered by BM-CL are not substantial. While it often reduces the leveling-down effect (LDE), this sometimes comes at the cost of lower worst-group improvement compared to simpler baselines (e.g., compare GroupDRO-LwF vs GroupDRO on CheXpert). The improvements in balancing the trade-off are marginal and may not be significant enough to justify the method's adoption.

3. Unjustified Complexity vs. Benefit: The BM-CL framework introduces significant complexity compared to standard ERM or single-stage bias mitigation methods. It requires a two-stage pipeline, identification of best/worst groups, and tuning of additional CL-specific hyperparameters (p, λ). Given the limited novelty and marginal performance gains, the cost-benefit trade-off appears unfavorable. The paper does not sufficiently demonstrate that this added complexity yields a practically meaningful advantage.

**Questions:**

1.Can the authors articulate the core technical novelty of BM-CL beyond the combination of existing CL regularization techniques with existing bias mitigation losses? What is fundamentally new in the mechanism proposed?

2.Given that the empirical improvements over baselines appear marginal (Table 1), how do the authors justify the significant added complexity (two stages, extra hyperparameters) of the BM-CL framework from a practical cost-benefit perspective?

---

> ### Author Response · Authors · 2025-11-21
> **1/2**
>
> We thank the reviewer for highlighting that our paper “addresses a critical and practical problem in fairness”. In what follows, we discuss their comments:
>
> **W1 and Q1: Novelty and main contributions**
>
> We respectfully disagree with the reviewer in that our paper shows limited novelty or an incremental contribution. As also noted by the other reviews, the main novelty of our work is the introduction of a new framing that interprets the leveling-down effect as an instance of catastrophic forgetting. We emphasize that our contribution is not the introduction of yet another bias mitigation algorithm, but a new formulation that casts fairness as a continual learning problem. We note that time and time again novel fruitful practices in ML have arisen from transplanting existing tools and techniques from other subfields.  In particular, the connection between fairness and continual learning has not been explored in prior work and enables analyzing fairness degradation through the lens of the stability–plasticity trade-off. BM-CL is a general framework built around this perspective, rather than a single-purpose algorithm. By reframing fairness degradation as a form of catastrophic forgetting, we establish a formal link between two previously disconnected research areas and open the door for cross-pollination between the fairness and continual learning communities.
>
> Although EWC and LwF are well-established continual learning tools, their reinterpretation as mechanisms that explicitly constrain forgetting on advantaged groups is new. Importantly, we not only propose this conceptual framework, but also demonstrate its practical value: BM-CL effectively modulates the worst–best group trade-off across three datasets, outperforming or matching prior bias mitigation approaches. Overall, BM-CL is a flexible and modular framework (agnostic to the specific mitigation method) and this generality is a core part of the contribution.
>
> Finally, we would like to address the attention of the reviewer to the new section we have included in the revised manuscript Appendix titled “A Lagrangian View of Fairness–Forgetting Trade-offs",, where we introduce a theoretical analysis about the BM-CL framework through a lagrangian view. This section was partially created in response to the comment of reviewer VVyQ, who noted that our framework was actually lacking a theoretical interpretation.
>
> **W2. Empirical improvements**
>
> The goal of BM-CL is not maximizing worst-group accuracy at all costs, but to improve disadvantaged groups while preventing harm to advantaged ones. From the stability-plasticity viewpoint, aggressive  improvement on $G_{worst}$ can directly harm $G_{best}$. BM-CL controls this via the CL regularizer Thus, even when absolute gains appear modest, BM-CL consistently produces positive-sum improvements, which baseline methods often fail to achieve.

---

> > ### Author Response · Authors · 2025-11-21
> > **2/2**
> >
> > **W3 and Q2. Complexity and benefit**
> >
> > Although BM-CL introduces an additional stage and one extra hyperparameter, the practical overhead is minimal. This minimal additional complexity was in fact also highlighted by Reviewer 2D32. The second stage reuses the same dataset as Stage 1, no new data or architectures are needed. Only one stability parameter $\lambda$ is added, and it follows a clear stability–plasticity interpretation. Both EWC and LwF are efficient in this single-transition setting (no sequence of tasks or replay buffer required). BM-CL provides a favorable cost–benefit ratio, as modest numerical gains, but consistent reduction of leveling-down, which baselines do not guarantee.
> >
> > BM-CL is designed to be simple to integrate into existing fairness pipelines. It requires no changes in model architecture, datasets or training pipeline. The framework is modular, since bias mitigation is handled by the chosen baseline (e.g., GroupDRO), while CL regularization ensures performance preservation. This separation of roles makes BM-CL easy to adopt in practice.
> >
> > We also note that the leveling-down effect, which is prevalent in many bias mitigation strategies, results in ethical conflicts in applications such as medical image computing. So gains that may seem small in other contexts may resolve important conflicts.
> >
> > **Main changes in the revised manuscript:** please note that we have uploaded a revised manuscript introducing changes based on your comments as well as the comments of the other reviewers. The new content is highlighted in violet. As a result, we have now included 1) a new benchmark method for bias mitigation (Invariant Risk Minimization), 2) a new section where we provide a more rigorous theoretical interpretation of our framework through a Lagrangian view of the fairness-forgetting trade off and 3) experiments on a new dataset of tabular data, to show that our framework can be applied to data modalities beyond images. We believe this revised manuscript is a much stronger version of our work. Should you agree that the revision improves the paper, we would appreciate your consideration in updating the score.

---

### Official Review · Reviewer_VVyQ · 2025-10-30

**Soundness:** 3
**Presentation:** 3
**Contribution:** 2
**Rating:** 6
**Confidence:** 3

**Summary:**

This paper introduces BM-CL, a novel framework that addresses the "leveling-down effect" commonly found in bias mitigation techniques. The core contribution is the reinterpretation of bias mitigation as a task-incremental continual learning problem. Experiments demonstrate that, compared to baseline methods, this approach effectively improves worst-group accuracy while significantly minimizing the drop in best-group performance , promoting positive-sum fairness without designing new complex architectures.

**Strengths:**

1. It cleverly utilizes existing CL techniques in a novel way without requiring additional datasets or complex model architectures, which helps promote positive-sum fairness.
2. The paper reinterprets the bias mitigation problem as a task-incremental continual learning problem. This is a highly novel perspective that offers a new approach to addressing the issue.
3. While traditional bias mitigation methods often sacrifice the performance of advantaged groups, this framework leverages CL principles to improve outcomes for disadvantaged groups while effectively preserving knowledge for advantaged groups, thereby avoiding performance degradation.

**Weaknesses:**

1. Although the conceptual framework is innovative, the novelty of the contribution is limited. It primarily involves adapting existing methods to the problem posed by the paper, rather than proposing an entirely new algorithm to solve it.
2. The method assumes that explicit group labels are available during training, which may limit its broader applicability.
3. The paper does not provide theoretical guarantees for fairness convergence. The proposed loss function is effectively a combination of two conflicting objectives. While the paper empirically demonstrates that a good balance point can be found, it does not prove that such a balance point must exist or can always be achieved. Furthermore, regarding the leveling-down boundary, although the experimental results show low leveling-down values, there is a lack of theoretical analysis to indicate how severe the worst-case leveling-down might be on more complex datasets.

**Questions:**

1. The ablation study indicates that the regularization strength $\lambda$ is a critical hyperparameter. Beyond conducting a grid search on the validation set, is there a more principled or intuitive method for setting this parameter?
2. In Table 1, you compare BM-CL against ERM, GroupDRO, JTT, and other methods. Could you include comparisons against more recent, SOTA bias mitigation algorithms as baselines? Including these comparisons would allow for a more comprehensive evaluation of the BM-CL framework's effectiveness.

---

> ### Author Response · Authors · 2025-11-21
>
> We thank the reviewer for highlighting that our paper “cleverly utilizes existing CL techniques in a novel way” and “is a highly novel perspective that offers a new approach” to address fairness issues.
>
> **W1. Novelty and contributions**
>
> The novelty of our work lies in the conceptual redefinition of fairness as a continual learning problem, rather than designing a completely new algorithm. We believe that this is a powerful idea, as this encourages the fairness community to benefit from the use of existing CL tools to address fairness-accuracy trade-offs, an aspect that, to our knowledge, has not been previously explored. BM-CL thus provides a contribution at framework-level, complementing existing bias-mitigation losses and generalizing across datasets and mitigation strategies, opening new joint avenues of research between the CL and fairness communities, both present at ICLR. Moreover, we evaluate our framework in multiple datasets and benchmark it against state of the art methods, confirming that it is useful in practice.
>
> **W2. Availability of group labels during training**
>
> We agree that requiring group labels may limit applicability in scenarios where such labels are unavailable. However, this assumption is standard in group-based fairness evaluation and in the benchmark datasets used in prior work. Our goal is not to replace label-free approaches, but to show that stability-based regularization can substantially mitigate leveling-down when group information is available.
>
> **W3. Theoretical analysis**
>
> We agree with the reviewer that the previous version of the manuscript lacked a deeper theoretical justification. In response, we have added a new section (Appendix C in the revised manuscript) that develops a Lagrangian view of the fairness-forgetting trade-off. Specifically, we reinterpret the formulation in Eq. 2 as a Lagrangian relaxation of a constrained optimization problem that explicitly limits forgetting on initially advantaged groups. This perspective enables a stability analysis that yields a bounded-harm guarantee for BM-CL: any improvement in the bias-mitigation loss can induce at most an increase in the continual-learning loss (i.e., deviation from the stage-one model) scaled by $\frac{1}{ \lambda}$. This directly explains why BM-CL effectively constrains leveling-down.
>
> **Q1. Intuitive method to set the regularization strength**
>
> The regularization strength $\lambda$ can be interpreted intuitively through the classical stability-plasticity trade-off. The CL term $L_{CL}$ penalizes deviations from the stage-1 model, which performs well on advantaged groups ($G_{best}$). This is conducted via distillation in LwF or via resistance to parameter drift measured by the Fisher information in EWC. Thus, a large $\lambda$ prioritizes stability, which means a strong preservation of performance on $G_{best}$ but limited improvements on $G_{worst}$. On the other hand, a small $\lambda$ indicates more plasticity, where the model can adapt more to $G_{worst}$, but may degrade performance for $G_{best}$. From this perspective, $\lambda$ controls the acceptable deviation from stage-1 predictions, so the optimal value is the smallest $\lambda$ that prevents accuracy degradation on $G_{best}$ while still enabling progress on $G_{worst}$. We have now included this discussion in Section 2.2.2 of the revised manuscript.
>
> **Q2. Comparison with additional bias mitigation strategies**
>
> Following the reviewer suggestion,  we have now included Invariant Risk Minimization (IRM) (Arjovsky et al., 2019), which is a widely used SOTA method in the bias-mitigation literature. IRM performs better in the best-group compared to other bias mitigation methods, but BM-CL remains superior (see Table 1 in the revised manuscript).
>
> **Main changes in the revised manuscript:** please note that we have uploaded a revised manuscript introducing changes based on your comments as well as the comments of the other reviewers. The new content is highlighted in violet. As a result, we have now included 1) a new benchmark method for bias mitigation (Invariant Risk Minimization), 2) a new section where we provide a more rigorous theoretical interpretation of our framework through a Lagrangian view of the fairness-forgetting trade off and 3) experiments on a new dataset of tabular data, to show that our framework can be applied to data modalities beyond images. We believe this revised manuscript is a much stronger version of our work. Should you agree that the revision improves the paper, we would appreciate your consideration in updating the score.

---

> > ### Comment · Reviewer_VVyQ · 2025-11-27
> >
> > I would like to thank the authors for their detailed response and the significant effort put into the revised manuscript. I have carefully reviewed the new content, particularly the Lagrangian view of the fairness-forgetting trade-off in Appendix C and the additional experiments on tabular data.
> >
> > 1. Acknowledging the Rebuttal Effort:
> >
> > The inclusion of the theoretical analysis addresses my previous concern regarding the lack of formal justification1. Additionally, the comparison with Invariant Risk Minimization (IRM) helps contextualize the method's performance against a broader range of baselines2.
> >
> > 2. On Contribution and Performance:
> >
> > However, regarding the core contribution, I maintain that the novelty is somewhat limited as the method primarily combines existing components (bias mitigation losses with CL regularization). The empirical gains in worst-group accuracy are present but often marginal, and the stability improvements, while consistent, do not represent a massive leap in performance.
> >
> > 3. Practical Value :
> >
> > That said, I see a distinct practical merit in this work that justifies its acceptance. As supported by the new theoretical analysis, the CL regularization effectively provides a "bounded-harm guarantee"3333. In practical applications, a method that ensures the result "does not get worse" (i.e., strictly limiting the degradation of the advantaged group) is often more valuable than a method that aggressively chases fairness scores at the risk of destabilizing the model's primary utility. The ability to control this trade-off via $\lambda$ offers a necessary safety net for deployment4.
> >
> > Conclusion:
> >
> > Taking these factors into account, I consider the quality of this paper to be slightly above the standard acceptance threshold. It offers a solid, safe, and logical framework for bias mitigation. However, considering the incremental nature of the technical novelty and the overall consensus on the magnitude of the improvements, I will maintain my current score of 6.

---

### Official Review · Reviewer_3nuT · 2025-10-30

**Soundness:** 3
**Presentation:** 4
**Contribution:** 3
**Rating:** 6
**Confidence:** 5

**Summary:**

This paper introduces Bias Mitigation through Continual Learning (BM-CL), a novel framework that reinterprets the challenge of algorithmic fairness as a domain-incremental continual learning problem. The key contribution is a methodology that leverages techniques like Learning without Forgetting and Elastic Weight Consolidation to incrementally adapt models to new fairness objectives. This approach specifically mitigates the common leveling-down effect by improving outcomes for disadvantaged groups without degrading performance for advantaged groups, thereby balancing fairness with the preservation of the model’s original knowledge, as validated on diverse synthetic and real-world datasets.

**Strengths:**

S1: The primary strength of this work is its novel conceptual contribution of bridging the fields of fairness and continual learning.

S2: By reinterpreting bias mitigation as a domain-incremental or task-incremental continual learning problem, this work leverages techniques like Learning without Forgetting and Elastic Weight Consolidation to incrementally adapt models to new fairness objectives.

S3: The effectiveness of BM-CL is validated across multiple, diverse real-world image datasets such as Waterbirds, CelebA, and CheXpert.

**Weaknesses:**

W1: Methods such as EWC and LwF can struggle with scalability to a very large number of fairness tasks or domains, and their effectiveness can be sensitive to hyperparameter tuning.

W2: The experiments are confined to image classification tasks. It remains unclear how well the BM-CL framework would generalize to other data modalities, such as tabular data or natural language, where biases are equally prevalent and challenging.

W3: The work focuses on group fairness metrics and accuracy but may not have extensively explored other important fairness notions like individual fairness or the long-term societal impact of deploying such models.

**Questions:**

- How exactly is the bias mitigation task structured as a continual learning problem? Is the model first trained on the original biased dataset and then fine-tuned on a de-biased or re-weighted version? Or is it exposed to different demographic groups sequentially?
- How sensitive are the results to the order in which the fairness conditions or groups are introduced? Continual learning performance is often highly dependent on task sequence.
- For the Elastic Weight Consolidation (EWC) method, how were the Fisher Information Matrix, the pretraining ratio, and the importance weight (lambda) for each parameter determined? Was lambda tuned specifically for fairness, and if so, what was the objective?
- For Learning without Forgetting (LwF), how were the soft targets from the previous model obtained and used? Was the original training data required for replay, or was it done solely with new data? If the latter, how does this affect the stability-plasticity balance?
- Were the hyperparameters (especially the critical ones for LwF and EWC like distillation temperature, regularization strength) optimized separately for each method, or was a consistent framework used? A difference in tuning effort could explain performance gaps.
- Beyond standard bias mitigation techniques, were other continual learning methods (e.g., experience replay with a small memory buffer) tested for comparison to better isolate the contribution of LwF/EWC?

---

> ### Author Response · Authors · 2025-11-21
> **1/2**
>
> We thank the reviewer for highlighting that “the primary strength of this work is its novel conceptual contribution”, and appreciating that our method is “validated across multiple, diverse real-world image datasets such as Waterbirds, CelebA, and CheXpert”. In what follows, we address the main points raised by the reviewer:
>
> **W1. Scalability of BM-CL**
>
> In BM-CL the task does not change: both stages operate on the same dataset and same prediction objective. The only difference lies in the optimization goal (ERM vs. Fairness + CL), not in the data distribution of the number of tasks. Moreover, the stability regularizer acts only on $G_{best}$, not on all groups, thus, the computational cost depends on the size of $G_{best}$, not on the number of demographic groups. For reducing computational overhead, in the EWC, the Fisher information could be even estimated using only a subset of $G_{best}$ samples.
>
> **W2. BM-CL in other data modalities beyond images**
>
> Our focus on image datasets was motivated by the presence of spurious correlations and demographic imbalances in vision benchmarks, which makes them suitable for fairness. However, the BM-CL formulation is modality-agnostic since it only assumes a base model, a fairness objective and a stability regularizer. We have now included experiments on tabular data using the Adult dataset, defining groups by race, sex, and income level. Results show that BM-CL achieves more favorable trade-offs between mitigation and limiting leveling-down, consistent with the trends observed in the image datasets (see Table 1 in the manuscript).
>
> **W3. BM-CL and other notions of fairness**
>
> We acknowledge that individual fairness and long-term societal effects are important dimensions of fairness. However, our work focuses on group fairness because it is the dominant paradigm in the datasets we study and aligns with prior literature in bias mitigation. We have added a brief remark in the conclusion about the relevance of this topic, leaving the door open for future avenues of research where BM-CL is used to address other fairness notions.

---

> > ### Author Response · Authors · 2025-11-21
> > **2/2**
> >
> > **Q1. How exactly is the bias mitigation task structured as a continual learning problem?**
> >
> > BM-CL follows a two-stage formulation. In stage 1, we train an ERM model (without fairness objectives). This naturally induces advantaged ($G_{best}$) and disadvantaged ($G_{worst}$) groups. In stage 2, we fine-tune the model **on the same data**, but now optimizing a bias-mitigation objective (Group-DRO or ReSample) that prioritizes $G_{worst}$, with a CL regularizer (EWC or LwF) that anchors the model to its behaviour in stage 1 on $G_{best}$. This makes fairness adaptation analogous to learning a “new task” while retaining knowledge from the previous one.
> >
> > **Q2. Dependence on group order**
> >
> > Sorry for the misunderstanding. We would like to clarify that BM-CL does not alternate between datasets or subgroups, both stages use the same training data. What changes is the objective, not the data order. In stage 1, we optimize for overall accuracy (ERM), while stage 2 optimizes bias mitigation with a stability regularizer. Because we do not introduce multiple sequential tasks derived from different groups, classical order-sensitivity issues existing in CL do not arise here.
> >
> > **Q3. EWC details and parameters**
> >
> > For EWC, we compute the empirical version of the Fisher information using samples from $G_{best}$, which takes the gradient of the loss using the true labels. The pretraining ratio $\rho$ was tuned via grid search over $\\{0.1, 0.2, 0.3\\}$, maximizing the worst-group accuracy on the validation set. The same procedure was used for the regularization strength $\lambda$ over $\\{0.0, 0.1, 1.0, 10.0\\}$. This protocol for tuning hyperparameters was applied across methods and datasets. These aspects are now clarified in Section 2.2.2 and Section 3.2 of the revised manuscript.
> >
> > **Q4. LwF details**
> >
> > For LwF, the soft-targets are obtained after stage 1 by applying the ERM model to $G_{best}$ samples. Specifically, we scale logits by $1/{T}$, where $T$ is the temperature factor, and compute probabilities using softmax. In our experiments, we set $T=2$ as it provides a well-established balance between softening the output distribution and maintaining stable gradients during distillation. During stage 2, the CL term is the KL divergence between these soft-targets and the current predictions on $G_{best}$. Importantly, BM-CL does not require replay mechanisms because both stages operate on the same dataset, which simplifies the stability-plasticity trade-off of classical CL.
> >
> > **Q5. Hyper-parameter tunning for LwF and EWC**
> >
> > For LwF, the temperature $T$ is fixed across all experiments. For both LwF and EWC, the regularization strength \lambda is tuned using the same grid search, targeting the worst-group accuracy on the validation set (see Fig. 3). As discussed in Section 4.2, EWC values are multiplied by $10^3$ due to the scale of the Fisher penalty.
> >
> > **Q6. Inclusion of other CL methods**
> >
> > We did not evaluate additional CL methods beyond EWC and LwF. We selected these two because they are widely used and conceptually aligned with our goal of preserving prior performance for specific groups. We have now added a short discussion of these directions in the conclusion section of the revised manuscript.
> >
> > **Main changes in the revised manuscript:** please note that we have uploaded a revised manuscript introducing changes based on your comments as well as the comments of the other reviewers. The new content is highlighted in violet. As a result, we have now included 1) a new benchmark method for bias mitigation (Invariant Risk Minimization), 2) a new section where we provide a more rigorous theoretical interpretation of our framework through a Lagrangian view of the fairness-forgetting trade off and 3) experiments on a new dataset of tabular data, to show that our framework can be applied to data modalities beyond images. We believe this revised manuscript is a much stronger version of our work. Should you agree that the revision improves the paper, we would appreciate your consideration in updating the score.

---

### Author Response · Authors · 2025-11-21
**Revised manuscript and response to reviewers**

We thank all reviewers for their comments and suggestions. We have revised the manuscript and uploaded a new version in which all modifications are highlighted in violet. Some of the main changes include: 1) a new benchmark method for bias mitigation (Invariant Risk Minimization), 2) a new section where we provide a more rigorous theoretical interpretation of our framework through a Lagrangian view of the fairness-forgetting trade off and 3) experiments on a new dataset of tabular data, to show that our framework can be applied to data modalities beyond images. We believe this revised manuscript is a much stronger version of our work, and invite the reviewers to update the scores accordingly in case you find it appropriate.

Below, we provide a detailed point-by-point response to each reviewer in their respective threads.

---

### Meta-Review · Area_Chair_17zn · 2025-12-17

**Summary:**

While the reviewers agree the paper bridges fairness and continual learning in an interesting way, there are also concerns on the technical depth/novelty, contribution, underlying theory, marginal empirical gains, and overhead. While two reviewers are satisfied with the rebuttal, the other two have not given any indication of increasing their scores. Looking at the rebuttals, the novelty and empirical improvements seem debatable and should have been made clear in the paper itself in the first place and not just be discussed in the rebuttal. Even the revised paper does not seem to reflect these discussions, which are critical. While the authors made significant efforts in adding content in the revision, the fact that many changes were made means that the revision needs to be carefully examined again. Overall, it is difficult for the AC to champion this paper based on the current rebuttal.

**Reviewer Concerns:**

While two reviewers are satisfied with the rebuttal, the other two may still have concerns on the novelty, contribution, empirical gains, and overhead.

**Reviewer Scores:**

Overall I do not see much indication of the scores changing.
* Reviewer 3nuT (score: 6) seems satisfied with the rebuttal.
* Reviewer VVyQ (score: 6) stated the score will not change and is intended to be slightly above acceptance.
* Reviewer 23mH (score: 2) did not respond to the rebuttal.
* Reviewer 2D32 (score: 4) did not respond to the rebuttal.

---

### Decision · Program_Chairs · 2026-01-26

Reject